# The Pentose Phosphate Pathway: From Mechanisms to Implications for Gastrointestinal Cancers

**DOI:** 10.3390/ijms26020610

**Published:** 2025-01-13

**Authors:** Jincheng Qiao, Zhengchen Yu, Han Zhou, Wankun Wang, Hao Wu, Jun Ye

**Affiliations:** 1Department of Gastroenterology, The Second Affiliated Hospital, Zhejiang University School of Medicine, Hangzhou 310009, China; 22218066@zju.edu.cn (J.Q.); 22418355@zju.edu.cn (Z.Y.); 2Cancer Institute (A Key Laboratory for Cancer Prevention & Intervention, China National Ministry of Education), The Second Affiliated Hospital, Zhejiang University School of Medicine, Hangzhou 310009, China; 22418396@zju.edu.cn; 3Department of Surgical Oncology, The First Affiliated Hospital, Zhejiang University School of Medicine, Hangzhou 310016, China; wkwang@zju.edu.cn

**Keywords:** pentose phosphate pathway, gastrointestinal cancers, redox homeostasis, nucleotide biosynthesis, tumor microenvironment

## Abstract

The pentose phosphate pathway (PPP), traditionally recognized for its role in generating nicotinamide adenine dinucleotide phosphate (NADPH) and ribose-5-phosphate (R5P), has emerged as a critical metabolic hub with involvements in various gastrointestinal (GI) cancers. The PPP plays crucial roles in the initiation, development, and tumor microenvironment (TME) of GI cancers by modulating redox homeostasis and providing precursors for nucleotide biosynthesis. Targeting PPP enzymes and their regulatory axis has been a potential strategy in anti-GI cancer therapies. In this review, we summarize the regulatory mechanisms of PPP enzymes, elucidate the relationships between the PPP and TME’s elements, and discuss the therapeutic potential of targeting the PPP in GI cancers.

## 1. Introduction

Gastrointestinal (GI) cancers are prevalent and have a detrimental impact on patients’ quality of life and their capacity for productivity, leading to substantial healthcare utilization [1]. According to the global cancer statistics by world region for the year 2022, GI cancers account for nearly one-quarter of the global cancer incidence and one-third of all cancer-related deaths [2].

Metabolic reprogramming, a hallmark of cancer, is described as alterations in key metabolic pathways to meet the metabolic needs of tumor cells [3]. The Warburg effect refers to the preference of cancer cells to convert glucose to lactate through glycolysis, even in the presence of sufficient oxygen and functional mitochondria, which is closely linked to rapid cancer cell proliferation [4]. This effect not only enhances cellular energy production and generates reducing power but also provides carbon precursors for the synthesis of nucleotides, lipids, and amino acids, thus promoting cancer progression [5]. As a glucose-oxidizing pathway that runs in parallel to upper glycolysis, the role of the pentose phosphate pathway (PPP) in the occurrence and development of malignant cancers is receiving more attention [6,7,8,9].

Briefly, the PPP is divided into two branches: the oxidative branch (oxPPP) and the non-oxidative branch (non-oxPPP). The oxPPP generates nicotinamide adenine dinucleotide phosphate (NADPH), a crucial electron donor for biosynthetic reactions and a protector against oxidative stress [10]. The non-oxPPP contributes to the production of ribose-5-phosphate (R5P), a precursor for nucleotide biosynthesis [11]. Thus, it integrates the biosynthetic demands with the maintenance of redox homeostasis, playing a pivotal role in supporting anabolic growth and protecting cells from oxidative stress [12]. During this process, glucose-6-phosphate dehydrogenase (G6PD) is the rate-limiting enzyme of the PPP, while transketolase (TKT) is the key enzyme in the non-oxPPP. A summary of the metabolic flux of the PPP is shown in Figure 1. However, whether and how the key enzymes or flux of the PPP function in GI cancer initiation and progression remain to be explored further.

In this review, we summarize the regulatory mechanisms of critical enzymes in the PPP during GI cancer pathogenesis. We also discuss the relationships between the PPP and elements of the tumor microenvironment (TME) in GI cancers. Furthermore, the role of the PPP in current pharmaceutical agents and therapeutic strategies for GI cancer treatment is highlighted.

## 2. Regulatory Mechanisms of the PPP in GI Cancers

The PPP plays an important role in GI cancers. Generally, enzymes like G6PD and TKT regulate cell proliferation, invasion, and metastasis, thereby promoting tumorigenesis, progression, and treatment resistance. Abnormal activation of the PPP is related to genetic mutations, further driving cancer progression [13]. Additionally, the PPP generates NADPH necessary for redox homeostasis and influences inflammation progression, which is also related to the development of GI cancers [14]. The multifaceted regulatory mechanisms of the PPP in various GI cancers are elucidated as follows.

### 2.1. Esophageal Cancer

In esophageal squamous cell carcinoma (ESCC), G6PD is an independent prognostic factor [15]. Polo-like kinase 1 (PLK1) coordinates biosynthesis during cell cycle progression and promotes cancer cell growth by directly increasing G6PD phosphorylation and activating the PPP [16]. Research has also established that the knock-out of PLK1 promoted ferroptosis through the inhibition of the PPP in ESCC [17]. In rat duodenal reflux models, chronic bile acid exposure triggers G6PD overexpression and nuclear factor kappa B (NF-κB) activation, potentially inducing genetic mutations and facilitating ESCC progression [13]. DNA polymerase ι promotes ESCC proliferation by activating G6PD and shunting glucose flux towards the PPP through O-GlcNAc transferase (OGT)-promoted O-GlcNAcylation [18].

Transketolase-like-1 (TKTL1), one of the three isoforms of TKT, is involved in the regulation of multiple cancer-related events [19]. TKTL1 overexpression is linked to the heightened aggressiveness of ESCC. Specifically, *TKTL1* expression is positively correlated with the expression of several cell proliferation-related genes and pro-metastasis genes and is negatively correlated with the expression of apoptosis-related genes from the BCL-2 family and anti-metastasis genes [20]. In addition, Liu et al. found that high-mobility group AT-hook 1 (HMGA1), a structural transcriptional factor, upregulates the expression of TKT by enhancing the binding of specificity protein 1 (Sp1) to *TKT* promoter, thereby promoting ESCC tumorigenesis [21].

### 2.2. Gastric Cancer

In gastric cancer (GC), LINC00242 competitively combined miR-1-3p, thus relieving miR-1-3p-mediated suppression on *G6PD* and promoting aerobic glycolysis and GC progression [22]. As a nuclear receptor, Rev-erbα binds to the promoters of the *6-phosphofructokinase-2/fructose-2,6-bisphosphatase 3* (PFKFB3) and *G6PD* genes, thereby inhibiting their transcription. Consequently, reduced expressions of Rev-erbα significantly increase the proliferation of GC and are positively correlated with the advanced TMN stage and poorer prognosis of patients [23].

TKTL1 is a biomarker for the poor prognosis of GC patients, and an elevated expression of TKTL1 in patients is associated with reduced chemosensitivity to docetaxel, oxaliplatin, and S-1 [24,25,26]. In addition, with shRNA silencing *TKTL1* in GC cells, their proliferation and tumor growth were inhibited, while the cell cycle was delayed in the G_0_/G_1_ phase [27].

### 2.3. Colorectal Cancer

In colorectal cancer (CRC), G6PD maintains redox balance and shields cancer cells from oxidative stress, contributing to cancer progression [28]. Zhang et al. highlighted the ability of p21-activated kinase 4 (PAK4) to boost G6PD activity through enhanced murine double minute 2 (MDM2)-mediated p53 ubiquitination degradation [29]. Furthermore, G6PD phosphorylation, controlled by neuronal differentiation 1 (NeuroD1) and tyrosine kinase c-Src, activates the PPP [30,31]. The Ras-related C3 botulinum toxin substrate 1 (Rac1) upregulates the expression of the sex-determining region Y-box 9 (SOX9) through the PI3K/AKT signaling pathway. SOX9 directly binds to the promoters of *Hexokinase 2* (HK2) and *G6PD*, enhancing their transcription. This increases glycolysis and promotes the PPP, ultimately promoting the proliferation, invasion, and migration of CRC [32]. Pre-B-cell leukemia transcription factor 3 (PBX3) has been found to bind directly to the *G6PD* promoter, leading to PPP stimulation and enhancing the production of nucleotides and NADPH [33]. Otherwise, circNOLC1 interacts with AZGP1 under Yin Yang 1 (YY1) regulation to activate the mTOR/SREBP1 signaling pathway, resulting in c-Met overexpression and G6PD activation [34]. Interestingly, YY1 plays multifaceted roles in CRC. First, the YY1/ELFN1-AS1/TP53/G6PD axis is identified as a regulator axis of G6PD, linking its oncogenic activity with tumor cell metabolic reprogramming [35]. Second, another YY1-mediated regulation of the PPP in CRC is not through p53 but rather through the direct activation of G6PD transcription by YY1 [36]. Lastly, circAGFG1 was found to drive metastasis and stemness in CRC by modulating the YY1/CTNNB1 axis, enhancing G6PD activity [37]. Furthermore, the overexpression of ATPase cation transporting 13A2 (ATP13A2) not only enhances the nuclear localization of transcription factor EB (TFEB) but also inhibits its phosphorylation, leading to an increased expression of 6-phosphogluconate dehydrogenase (6PGD) and elevated activity of the PPP [38].

TKT interacts with glucose-regulated protein 78 (GRP78) and promotes CRC metastasis by regulating Akt phosphorylation [39]. Sirtuin 5 (SIRT5), a member of the NAD^+^-dependent class III histone deacetylase family, directly interacts with TKT, leading to its demalonylation and activation, which enhances the PPP and DNA replication, ultimately delivering poor prognosis [40]. Additionally, the high expression of TKT is linked to the upregulation of the Notch signaling pathway, thereby enhancing the proliferation and migration of CRC cells [41]. Elevated TKTL1 expression in CRC enhances glucose metabolism independent of oxygen and matrix degradation, which is fueled by lactate production, thus promoting metastasis [42]. When oncogenic signals suppress p16, it leads to uninhibited cell cycle progression and allows cells to bypass oncogene-induced senescence. p16 suppression activates the mechanistic target of rapamycin complex 1 (mTORC1) signaling, which in turn increases the expression of ribose-5-phosphate isomerase A (RPIA), a key enzyme in the PPP. The upregulation of RPIA enhances nucleotide synthesis, leading to an increased production of deoxyribonucleotides, which are essential for DNA replication and proliferation [43]. Importantly, RPIA transcends its traditional enzymatic role by entering the nucleus to form a complex with adenomatous polyposis coli (APC) and β-catenin. This interaction prevents the phosphorylation, ubiquitination, and degradation of β-catenin, leading to its upregulation [44]. This process is important as the aberrant activation of the Wnt/β-catenin signaling pathway is a common oncogenic event in CRC. Persistent Wnt/β-catenin signaling facilitates the epithelial–mesenchymal transition (EMT), accelerating CRC invasion and metastasis [45]. Moreover, the TP53-induced glycolysis and apoptosis regulator (TIGAR) primarily reduces glycolysis and increases PPP flux, thereby boosting the production of R5P and NADPH, which are important for intestinal regeneration and are associated with the prevalence of CRC [46,47].

### 2.4. Liver Cancer

In liver cancer, elevated G6PD expression is significantly associated with metastasis and poor prognosis of hepatocellular carcinoma (HCC) in patients. Mechanically, G6PD promotes EMT by upregulating the STAT3 signaling pathway, which ultimately enhances migration and invasion in HCC [48]. In a rat model of hepatocarcinogenesis, where early preneoplastic foci and nodules progressed towards HCC, metabolic changes were found to be characterized by enhanced activity of the PPP and reduced oxidative phosphorylation (OXPHOS) through the upregulation of nuclear factor erythroid 2-related factor 2 (NRF2), leading to the overexpression of G6PD [49]. Interestingly, NRF2 activation is also evident in hepatitis B virus (HBV) infection. In hepatocytes, HBV stimulates G6PD expression through HBV X protein (HBx) in an NRF2 activation-dependent pathway. HBx binds to the UBA and PB1 domains of the adaptor protein p62, enhancing the interaction between p62 and the NRF2 repressor kelch-like ECH-associated protein 1 (Keap1), forming an HBx–p62–Keap1 complex in the cytoplasm. This complex sequesters Keap1, reducing its inhibition of NRF2, leading to NRF2 activation and, subsequently, increasing G6PD transcription [50]. Another study confirmed that RNA interference targeting *G6PD* significantly reduced HBV replication, decreasing it by fivefold via the IFN pathway [51]. This indicates that the mechanism related to the NRF2/G6PD axis plays an important role in HBV-related liver cancer development and progression. In HCC, phosphatase and tensin homolog located on chromosome 10 (PTEN) has also been reported to negatively regulate G6PD through three distinct mechanisms. First, PTEN activates glycogen synthase kinase-3β (GSK3β), which phosphorylates T cell leukemia 1 (Tcl1), thereby inhibiting Tcl1’s interaction with heterogeneous nuclear ribonucleoprotein K (hnRNPK). Second, PTEN binds to hnRNPK, preventing the cleavage of *G6PD* pre-mRNA. Third, PTEN directly interacts with G6PD, inhibiting its dimerization and activity. These findings suggest that the PTEN/Tcl1/hnRNPK/G6PD axis could be a potential therapeutic target to improve the prognosis for HCC patients [52]. In HCC, aldolase B (Aldob) potentiates the p53-mediated inhibition of G6PD through the formation of an Aldob–G6PD–p53 complex, which is independent of Aldob enzymatic activity [53]. Furthermore, Bcl-2-associated athanogene 3 (BAG3) directly interacts with G6PD and suppresses PPP flux, de novo DNA synthesis, and cell growth in HCC [54]. Interestingly, some researchers have observed that although both G6PD and malic enzyme mRNA expression are increased, G6PD activity is inhibited in both hyperplastic liver and HCC by peroxisome proliferators, while malic enzyme activity is elevated to support NADPH production and cholesterol synthesis [55]. Additionally, the amplification of c-MYC has been observed in HCC patients, which drives cholesterol synthesis [56]. Hu et al. found that c-MYC promoted a positive feedback loop between cholesterol synthesis and the PPP, which drove the proliferation of malignant hepatocytes. In the meantime, mR-206 directly repressed the expression of *HMGCR* and *G6PD*, thereby disrupting the positive feedback loop [57]. Moreover, 6PGD has been identified as an independent prognostic factor for HCC patients, with high expression levels correlating with worse prognosis and beneficial efficacy of immunotherapy [58]. NRF2 is overexpressed in HCC and directly binds to the antioxidant response element in the *6PGD* promoter region, enhancing its expression. Increased 6PGD expression, in turn, upregulates NRF2, forming a positive feedback loop between NRF2 and 6PGD. This loop ultimately leads to increased cell proliferation, survival, and migration in HCC [59].

TKT has been identified as a driver of HCC development by counteracting oxidative stress through the NRF2/KEAP1/BTB and CNC homolog 1 (BACH1) pathway [60]. Except for its metabolic function, the nuclear localization of TKT has been demonstrated to promote HCC through the activation of the epidermal growth factor receptor (EGFR) pathway in a non-metabolic manner [61]. As a Ser-Thr kinase, vaccinia-related kinase 2 (VRK2) can promote Thr287 phosphorylation of TKT and then facilitate E3 ubiquitin ligase F-Box and Leucine-Rich Repeat Protein 6 (FBXL6)-mediated ubiquitination and activation of TKT. Activated TKT further upregulated programmed death-ligand 1 (PD-L1) and VRK2 expression by decreasing reactive oxygen species (ROS) accumulation and mTOR activation, resulting in immune evasion and HCC metastasis [62]. Moreover, Zheng et al. revealed an interaction between TKT and SH2 domain-containing 5 (SH2D5), induced by HBx, which promotes HCC cell proliferation [63]. The overexpression of RPIA in HCC leads to ERK phosphorylation, increased lipid synthesis, and the activation of the β-catenin signaling pathway, ultimately prom oting cancer progression [64]. Compared to normal hepatocytes, HCC cells have a significantly lower fructose metabolism rate and ROS level due to the expression of the high-activity ketohexokinase (KHK)-A isoform. As a protein kinase, KHK-A phosphorylates and activates phosphoribosyl pyrophosphate synthetase 1 (PRPS1) to promote PPP-dependent de novo nucleic acid synthesis and HCC formation [65].

### 2.5. Pancreatic Cancer

In pancreatic ductal adenocarcinoma (PDAC), patients with high glucose metabolism levels have a worse prognosis. Mechanically, these patients have high glucose transporter 1 (GLUT1) and low Aldob expression, leading to increased glycolytic flux, G6PD activity, and pyrimidine biosynthesis [66]. In addition, PDAC distant metastases have resulted in a core pentose conversion pathway, which converts glucose-derived metabolites into 6PGD substrate, thereby hyperactivating 6PGD to support tumor growth [67].

Histone H3 lysine 4 trimethylation (H3K4me3) protein was specifically recruited to the promoter of *TKT*, which is facilitated by S100 calcium-binding protein A11 (S100A11) interacting with SET and MYND domain-containing 3 (SMYD3), thus enhancing PPP flux and further promoting PDAC progression [68]. The short isoform of the prolactin receptor (PRLR) activates the Hippo pathway by interacting with NIMA-related kinase 9 (NEK9), thus inhibiting *G6PD* and *TKT* expression and contributing to proliferation inhibition in PDAC [69]. Moreover, oncogenic *KRAS* has been found to activate a MAPK-dependent signaling pathway, leading to MYC upregulation and transcription of the *RPIA*, which facilitates nucleotide biosynthesis to support PDAC growth [70]. The long non-coding RNA (lncRNA) growth arrest-specific 5 (GAS5) plays a vital role in the emergence of the CD133^+^ population, representing tumor-initiating cells that result in tumor relapse. In PDAC, the CD133^+^ cell population was found to redirect glucose to the PPP, which was predominantly biosynthetic. Despite being quiescent in nature, these cells did not use it immediately for nucleotide synthesis [71].

### 2.6. Other Cancers

In gastrointestinal stromal tumors (GISTs), when HIF-1α binds to the *6PGD* promoter sequence, it upregulates *6PGD* expression, leading to increased NADPH production. This rise in NADPH counteracts ROS-induced oxidative damage, stimulating GIST cells to progress from the G1 phase to the S phase of the cell cycle [72]. In cholangiocarcinoma, the overexpression of *G6PD* is also associated with decreased mitochondrial ROS and increased cisplatin resistance, which could be reversed by chloroquine via the inhibition of the autophagy lysosome pathway [73].

The regulatory axis and function of G6PD and other key enzymes in the PPP in GI cancers are shown in Figure 2 and Table 1.

## 3. The PPP and the TME in GI Cancers

The TME encompasses a range of cells, including immune cells, cancer-associated fibroblasts (CAFs), endothelial cells, pericytes, and other tissue-residing cells [79]. While previously seen as bystanders to tumorigenesis, these host cells are now recognized as key players in the development and advancement of cancer. The PPP plays a significant role in the dynamic interplay between tumor cells and the TME. Tumor cells within the TME can adapt to the intricate microenvironmental conditions by reprogramming the PPP. Therefore, exploring the relationships between the PPP and TME’s elements can provide insights into the crosstalk between tumor cells and the complex TME, which is summarized in Figure 3. 

### 3.1. The PPP and Nutrient Deprivation

The progression of tumor growth is accompanied by an enlargement in volume, potentially resulting in restricted nutritional support for the tumor cells [80]. Hence, the deficiency of glucose, serum, and amino acids is a characteristic of the TME.

Glucose deprivation enhances the binding of coactivator-associated arginine methyltransferase 1 (CARM1) and RPIA to induce the arginine 42 methylation of RPIA, thereby increasing the activity of RPIA and amplifying oxPPP flux, which contributes to the survival of CRC cells [74]. In addition, glucose deprivation and hydroxyethyl disulfide (HEDS) trigger p53-independent metabolic stress, including the loss of oxPPP function, thiol homeostasis, and sensitivity to radiation-induced oxidative stress in CRC cells [81]. Furthermore, PPP flux is increased when proline oxidase (POX) activity is enhanced due to glucose deprivation, leading to a boost in NADPH production. This increase in NADPH, in turn, contributes to the elevation of ATP levels through the proline cycle [82].

Numerous cancer cells rely upon glutamine, the most abundant amino acid in the blood, to replenish intermediates used for macromolecule biosynthesis [83]. Glutamine deprivation can increase the expression of G6PD through NRF2 activation, suggesting that targeting oxPPP enzymes and glutamine catabolism together is a strategy to combat CRC [84]. Moreover, De Falco et al. found that N-acetyltransferase 8-like (NAT8L) silenced HCC cells acquired proliferative advantage depending on glutamine oxidation. Specifically, the downregulation of NAT8L increases cytosolic aspartate levels to promote glucose flux into the PPP, thus boosting purine biosynthesis and ensuring HCC cell proliferation [85].

As for serum deprivation, it was found that macroH2A1-depleted HepG2 cells were insensitive to serum exhaustion. Specifically, the depletion of macroH2A1 in HCC cells leads to a significant enhancement of the PPP, which is crucial for providing precursors for nucleotide synthesis and supports the cancer stem cell-like metabolic phenotype [86].

LncRNAs also participate in the metabolic reprogramming of GI cancers under serum deprivation. For example, the stability and level of LINC01615 increase in a m^6^A-dependent manner under serum deprivation, thereby enhancing the expression of G6PD, the activation of PPP, and CRC cell survival [87].

### 3.2. The PPP and Hypoxia

Hypoxia within tumors arises due to the rapid proliferation of cancer cells and the insufficient balance between blood vessel development and oxygen availability. This state of low oxygen is a common, constant, and complex condition in the TME [88]. Facing the shift from normoxia to hypoxia, cells predominantly depend on the increased expression of hypoxia-inducible factors (HIFs) and the activation of HIF signaling pathways [89].

Singh et al. found that MUC1 physically interacts with HIF-1α and p300 in a hypoxia-dependent manner and facilitates the recruitment of HIF-1α and p300 on glycolytic gene promoters [90]. Consequently, PPP flux is enhanced by MUC1 overexpression, which contributes to nucleotide synthesis, resulting in radiation and gemcitabine resistance in PDAC [91,92]. In HCC, hypoxia is an extrinsic factor inducing phosphofructokinase-fructose bisphosphatase 4 (PFKFB4) expression in HCC in a HIF-1-dependent manner, which shifts the equilibrium from glycolysis to the PPP and alleviates cellular stress response, thus supporting HCC progression [93]. In addition, the nuclear translocation of TIGAR under genome stress or hypoxia increases, thereby activating the PPP and protecting HCC cells from DNA damage [94]. As a cell surface marker for cancer stem cells, CD44 ablation weakens the glycolytic phenotype of p53-deficient or hypoxic CRC cells and decreases metabolic flux to the PPP and glutathione (GSH) levels, which in turn increases the chemotherapy sensitivity of CRC cells [95].

### 3.3. The PPP and Acidosis

With a higher intracellular pH but a lower extracellular pH, the acidic niche is described as the acidosis of a tumor and its microenvironment, which is closely related to the hypoxia niche and lactate metabolism [96]. An acidic microenvironment redirects glucose away from lactate production and towards the oxPPP in order to produce NADPH and counter the increase in ROS present under acidosis. In terms of the mechanism, acidosis activates p53, which promotes the PPP, partly through the induction of G6PD expression [97]. Dichloroacetate, a kind of pyruvate dehydrogenase kinase (PDK) inhibitor, is more effective in decreasing the cell proliferation of acidic pH-adapted CRC cells compared to native CRC cells, which is related to a greater decrease in PPP activity [98]. In acidosis-adapted PDAC cells, PPP activity is enhanced compared to control cells, along with an increase in the inactivation of AMP-activated protein kinase (AMPK) and upregulation of matrix metalloproteinase-1 (MMP1), leading to higher proliferation, invasion, and metastasis ability [99].

### 3.4. The PPP and Tumor-Infiltrating Immunocytes

Throughout the stages of tumor progression, there is a dynamic and continuous interaction between tumor cells and the tumor-infiltrating immunocytes within the tumor immune microenvironment. These tumor-infiltrating immune cells include T lymphocytes, B lymphocytes, natural killer cells, neutrophils, tumor-associated macrophages, myeloid-derived suppressor cells, and dendritic cells. Each plays a distinct role in either promoting or inhibiting the progression of tumors within the microenvironment [100].

Within the inflammatory process, immune cells such as M1 macrophages and T-helper 17 cells undergo metabolic alterations characterized by an increased glucose uptake, reliance on glycolysis, and upregulated PPP. Conversely, cells with anti-inflammatory functions, like M2 macrophages, regulatory T cells, and quiescent memory T cells, exhibit decreased glycolytic activity and increased oxidative metabolism [101]. Modulating carbohydrate kinase-like (CARKL) intricately links the metabolic pathways and functional outcomes of M1 and M2 macrophages. When CARKL is downregulated, it decreases sedoheptulose-7-phosphate (S7P) and increases G6PD activity. This shift favors the pro-inflammatory M1 macrophage phenotype, enhancing the production of tumor necrosis factor α (TNFα) and interleukin-6 (IL-6), promoting oxPPP flux, and increasing NADPH production and ROS detoxification. Conversely, the upregulation of CARKL increases S7P levels, supporting the anti-inflammatory M2 macrophage phenotype [102].

Furthermore, the expression of PPP enzymes in immune cells is pivotal to their function. Tumor cells induce H3K9me3 deposition at the promoter of *G6PD*, resulting in decreased G6PD and granzyme B expression in tumor-specific cytotoxic T cells [103]. Although T cell proliferation depends on glycolysis, the differentiation of T effector cells requires ROS signaling. Genetic ablation or pharmacologic inhibition of the PPP enzyme 6PGD in the oxPPP promotes differentiation towards CD8^+^ T effector cells, enhancing tumoricidal activity and immunotherapy [104]. Considering that high levels of NADPH and GSH are essential for the formation and maintenance of CD8^+^ T memory cells, the cytosolic phosphoenolpyruvate carboxykinase (PCK1) increases glycogenesis to fuel PPP flux [105]. Additionally, a deficiency of TKT in regulatory T cells impairs their suppressive capability, induced by uncontrolled OXPHOS, lower α-ketoglutarate levels, and DNA hypermethylation [106].

### 3.5. The PPP and the Mechanical Microenvironment

The mechanical microenvironment within tumors, particularly concerning CAFs, plays a crucial role in shaping the TME and influencing tumor growth, metastasis, and resistance to therapy. CAFs significantly contribute to the deposition and remodeling of the extracellular matrix (ECM), altering the physical and biochemical TME in which cancer cells interact [107].

G6PD is highly expressed and activates the NF-κB signaling pathway, thereby promoting the production of hepatocyte growth factor (HGF) in gastric cancer-associated mesenchymal stem cells (GCMSCs). HGF, in turn, enhances GC cell proliferation and metastasis by upregulating HK2 [108]. In PDAC, the absence of focal adhesion kinase (FAK) in a subset of CAFs escalates tumor growth and boosts glycolysis in cancer cells due to the enrichment of the cytokine signaling pathway. Proteomics analysis has shown elevated levels of two key enzymes in the oxPPP, G6PD, and 6PGD in PDAC cells exposed to a FAK-depleted and CAF-conditioned medium [109]. As prominent stromal cells in PDAC, pancreatic stellate cells (PSCs) promote tumor progression through stromal-derived factor-1α (SDF-1α) and IL-6 secretion, which triggers cell proliferation via NRF2-mediated metabolic reprogramming and ROS detoxification. Moreover, G6PD downregulation or inhibition can attenuate this PSC-induced proliferation [77]. The loss of attachment to the ECM can initiate a range of cellular responses that influence cell survival. Detachment from the ECM triggers anoikis, a form of caspase-mediated apoptosis, characterized by significant metabolic disruptions. Upon ECM detachment, cells experience a notable reduction in ATP levels due to impaired glucose uptake, thus diminishing PPP flux and resulting in elevated ROS levels. The lack of glucose transport further compounds this metabolic impairment, which is crucial for ATP production through both glycolysis and the PPP. In matrix-detached cells, the downregulation of G6PD exacerbates oxidative stress by reducing NADPH production and increasing ROS accumulation. However, the overexpression of oncogenes like *ERBB2* can rescue these cells by stabilizing EGFR and activating the PI3K pathway, restoring glucose uptake and PPP flux, reducing ROS levels, and promoting cell survival through enhanced ATP generation and fatty acid oxidation [110]. The activation of serum and glucocorticoid kinase-1 (SGK1) is instrumental in anchorage-independent growth during ECM detachment. Mechanically, SGK1 activation promotes glucose uptake, PPP flux, glyceraldehyde-3-phosphate (G3P) production, and ATP generation, thereby enhancing CRC cell survival [111].

## 4. The Role of the PPP in Therapeutic Strategies for GI Cancers

The PPP is instrumental in influencing therapeutic outcomes in GI cancers, impacting chemoradiotherapy, targeted therapy, immunotherapy, combination therapy, and emerging therapy. Below, we will elaborate on these categories, detailing the underlying mechanisms of therapeutic strategies related to the PPP. Meanwhile, a detailed discussion of the impact of non-coding RNAs (ncRNAs) and therapeutic agents in the treatment of GI cancers is presented in Table 2 and Table 3, respectively.

### 4.1. Chemoradiotherapy

Personalized medicine is being used increasingly in cancer treatment, with a significant focus on discovering new biomarkers and therapeutic targets. Both chemotherapy and radiotherapy are now more precisely tailored to fit each patient’s unique profile. In ESCC, reduced expression of PLK1 can inhibit the PPP, leading to decreased levels of NADPH and GSH. This disruption triggers ferroptosis and enhances the sensitivity of cancer cells to paclitaxel, cisplatin, and radiotherapy [17]. Another study further demonstrated that silencing PKM2 increased cisplatin sensitivity in ESCC by inhibiting the PPP [112]. SET-domain-containing 2 (SETD2) deficiency leads to the upregulation of GLUT1 to meet the high glucose demand of PDAC cells. Concurrently, SETD2 deficiency directly inhibits the transcription of TKT, compromising nucleotide synthesis. The synergistic effect of these alterations increases the sensitivity of SETD2-deficient PDAC cells to gemcitabine when glycolysis is restricted [113]. In addition, PRLR reduces nucleotide synthesis through miRNA-induced G6PD and TKT inhibition, thus enhancing the sensitivity of PDAC cells to gemcitabine [114].

**Table 2 ijms-26-00610-t002:** Non-coding RNA related to the PPP in GI cancers.

GI Cancer Type	NcRNA	Role	Functions	Mechanisms	Reference
CRC	LINC01615	Oncogene	↑Survival, ↑nucleotide and lipid synthesis, ↓ROS production, ↑oxaliplatin resistance, and ↑PPP flux	Serum starvation/↓METTL3/↑LINC01615/competitive binding with hnRNPA1/↑G6PD	[87]
	CircNOLC1	Oncogene	↑Proliferation, ↑migration, ↑liver metastasis, and ↑PPP flux	YY1/↑CircNOLC1/AZGP1/↑mTOR/SREBP1 signaling/↑G6PD; YY1/↑CircNOLC1/↓miR-212-5p/↑c-Met/↑G6PD	[34]
	Circ_0003215	Tumor suppressor gene	↓Proliferation, ↓migration, ↓invasion, ↓metastasis, and ↓PPP flux	Circ_0003215/↓miR-663b/↑DLG4/↓G6PD	[115]
	ELFN1-AS1	Oncogene	↑Proliferation, ↑migration, ↑invasion, ↓apoptosis, and↑PPP flux	YY1/↑ELFN1-AS1/↓TP53/↑G6PD	[35]
	Lnc-AP	Tumor suppressor gene	↓Oxaliplatin resistance, ↑ROS accumulation, ↑apoptosis, and ↓PPP flux	Lnc-AP encoded pep-AP/↓TAL	[116]
	miR-124	Tumor suppressor gene	↓Growth, ↓nucleotide synthesis, and ↓PPP flux	miR-124/↓PRPS1 and RPIA.	[117]
HCC	miR-206	Tumor suppressor gene	↓Proliferation, ↓lipid accumulation, and ↓PPP flux	miR-206/↓G6PD	[118]
	miR-206	Tumor suppressor gene	↓Growth, ↓cholesterol synthesis, and ↓PPP flux	miR-206/↓G6PD and HMGCR	[57]
	miR-122, miR-1	Tumor suppressor gene	↓Viability and ↓PPP flux	miR-122 and miR-1/↓G6PD	[119]
PDAC	GAS5	Oncogene	↓Proliferation,↑quiescence, ↑metastasis, ↑invasion, and ↑PPP flux	Sox2/↑GAS5/↓glucocorticoid receptor transcriptional activity	[71]
	miR-4763-3p, miR-3663-5p	Tumor suppressor gene	↓Nucleotide synthesis, ↑gemcitabine sensitivity, and ↓PPP flux	PRLR/↑miR-4763-3p/↓G6PD PRLR/↑miR-3663-5p/↓TKT	[114]
GC	LINC00242	Oncogene	↑Aerobic glycolysis, ↑proliferation, ↓apoptosis, and ↓PPP flux	LINC00242/↓miR-1-3p/↑G6PD	[22]

“↑” means upregulation and “↓” means downregulation.

Research shows that inhibitor of differentiation 1 (ID1) promotes G6PD expression and PPP activation via the Wnt/β-catenin/c-MYC signaling pathway, contributing to oxaliplatin resistance and poor prognosis in patients [75]. Inhibiting G6PD reduces NADPH and GSH production, impairing their ability to clear ROS. This ROS-mediated damage subsequently enhances the apoptosis of CRC cells induced by oxaliplatin [28]. POU domain class 2 transcription factor 1 (POU2F1) binds directly to the aldolase A (ALDOA) promoter, thereby strengthening PPP activity in CRC. Silencing POU2F1, however, significantly increased CRC sensitivity to oxaliplatin [120]. Treatment with diallyl disulfide (DADS) disrupts the PPP, leading to decreased production of 5-phosphate ribose-1-pyrophosphate (PRPP). This results in increased DNA damage, enhanced cell apoptosis, and reduced growth of CRC cells, which is related to ubiquitination and degradation of POU2F1 [121].

As for radiotherapy, MUC1 expression reduced radiation-induced cytotoxicity and DNA damage in PDAC by promoting glycolysis, the PPP, and nucleotide biosynthesis. Pretreatment with the glycolysis inhibitor 3-bromopyruvate overcame MUC1-mediated radiation resistance, both in vitro and in vivo, by decreasing glucose entry into nucleotide synthesis pathways and increasing DNA damage [91]. At the same time, high TKT expression leads to radiotherapy resistance in clinical HCC patients, demonstrating its role in enhancing the auto-PARylation of poly (ADP-ribose) polymerase 1 (PARP1) in response to DNA double-strand breaks [122].

### 4.2. Targeted Therapy

Since their invention in the early 2000s, tyrosine kinase inhibitors (TKIs) have gained prominence as one of the most effective pathway-directed anti-cancer agents. Some TKIs have been taken to clinical trials in GI cancers, while imatinib is the mainstay of medical treatment of GISTs in the first-line metastatic setting [123,124].

PDAC resistance to the EGFR inhibitor erlotinib is closely linked to the upregulation of the PPP. Erlotinib-resistant PDAC cells show increased G6PD levels, reduced glycolytic activity, and lower glycolytic metabolites, while elevated G6PD levels are attributed to the upregulation of the ID1 [78]. Furthermore, erlotinib treatment could suppress the development of HCC by blocking the effect of TKT nuclear localization and inhibiting the EGFR pathway [61]. PTEN binds to G6PD in HCC, preventing the formation of active G6PD dimers, whereas Tcl1 can counteract this inhibition. Importantly, knocking down Tcl1 increases HCC sensitivity to sorafenib. This indicates that the PTEN–Tcl1 interaction plays a role in sorafenib resistance in HCC through the PPP [52]. Similarly, a positive feedback loop between the PPP and PI3K/AKT signaling upregulates G6PD activity, contributing to regorafenib resistance in HCC. Consequently, the PI3K/AKT pathway inhibitor MK-2206 or G6PD inhibitor 6-aminonicotinamide (6AN) can counteract this resistance [125]. In GISTs, resistance to imatinib is closely related to the PPP. Prolonged imatinib treatment increases intracellular ROS levels, which then causes an adaptive rise in HIF-1α protein levels. This rise in HIF-1α is accompanied by an increased expression of 6PGD, leading to the upregulation of the PPP. The upregulation of the PPP mitigates ROS-induced damage, promotes cell proliferation, and inhibits apoptosis, promoting the growth of resistant tumor cells [72].

As a novel TKT inhibitor, oroxylin A directly binds to TKT in HCC, decreasing its activity and expression. This results in the accumulation of non-oxPPP substrates and the activation of the p53 signaling pathway, ultimately inducing apoptosis and causing cell cycle arrest [126].

**Table 3 ijms-26-00610-t003:** Therapeutic agents related to the PPP in GI cancers.

GI Cancer Type	Agent	Target	Characteristics	Mechanisms	Reference
CRC	M4IDP	G6PD	Zoledronic acid derivative	↑Unprenylated Rap1A, RhoA and CDC42,↓G6PD,↑ROS, ↓NADPH and GSH, ↓mevalonate pathway, and ↓PPP flux	[127]
	Ankaferd hemostat	6PGD	Plant extracts of *Thymus vulgaris*, *Glycyrrhiza glabra*, *Vitis vinifera*, *Alpinia officinarum*, and *Urtica dioica*	↓6PGD,↑oxidative stress, and ↓PPP flux	[128]
	GO-203	MUC1 C-terminal subunit	D-amino acid cell-penetrating peptide	↓AKT-S6K1-elF4A pathway, ↓TIGAR,↓GSH/mitochondrial transmembrane potential, ↑ROS, and ↓PPP flux	[129]
	Piperlongumine/auranofin	Glutathione S-transferase π/Thioredoxin reductase	Natural alkaloid from piper longum L/trialkylphosphine gold complex	↑NRF2 target genes (G6PD), ↓CD44v9-positive fraction, ↓tumor formation and growth, and ↓PPP flux	[130]
	INK128/Avemar	mTOR	mTOR1/2 inhibitor/fermented wheat germ extract	↓PPP enzymes (G6PD, PGD, TKT), ↓NADPH/NADP+ and GSH/GSSG ratios, and ↓PPP flux	[131]
	Halofuginone	Akt/mTORC1 signaling	Derivative of the febrifugine	↓G6PD, ↑ROS, ↓NADPH, ↓glycolysis and lipid biosynthesis, and ↓PPP flux	[132]
	Epicatechin gallate	G6PD, TKT	Catechins in green tea and grape	↓Enzymatic activity of G6PD and TKT, ↓de novo synthesis of RNA ribose, and ↓PPP flux	[133]
	Aspirin	G6PD, TKT	Salicylate	↑Acetylation of G6PD and TKT,↓activity of G6PD, and ↓PPP flux	[134]
	Cu_2_O@Au nanocomposites	Ferroptosis pathway	Nanoparticles	↓GSH, ↑H_2_O_2_, ↑ferroptosis, ↑immune therapy response, and ↓PPP flux	[135]
	Phy@PLGdH nanosheets	6PGD	Nanoparticles	↓NADPH and nucleotide synthesis, ↑radiation-therapy-mediated oxidative stress and DNA damage, ↑immunogenic cell death, and ↓PPP flux	[136]
HCC	Oroxylin A	TKT	Herbal extracts	↓TKT activity, ↑non-oxPPP substrates, ↑p53 signaling, and ↓PPP flux	[126]
	CP-91149	Glycogen phosphorylase	Indole carboxamide	↑Effect of 6AN, ↑phosphorylation of AMPK, and ↓PPP flux	[137]
	Zerumbone	PI3K/AKT/mTOR and STAT3 signaling pathways	Sesquiterpene derived from the ginger plant zingiber zerumbet	↓Enzymes in PPP (G6PD, RPIA, RPE, TKT, and TAL), and ↓PPP flux	[138]
PDAC	Hypericin	G6PD	Naphthodianthrone, anthraquinone derivative, and active constituents of *Hypericum*	↓G6PD, ↓GSH, ↑ROS, ↑effect of gemcitabine, and ↓PPP flux	[139]
Cholangiocarcinoma	Chloroquine	Autophagy lysosome pathway	Antimalarials drug	↓G6PD, ↑mitochondrial ROS, ↑cisplatin-induced apoptosis, and ↓PPP flux	[73]

“↑” means upregulation and “↓” means downregulation.

### 4.3. Immunotherapy

Enzymes in the PPP participate in the survival, function, activation, and differentiation of T cells, which demonstrates their potential as a novel metabolic checkpoint for immunotherapy applications [103,104,140,141]. Dimethyl fumarate (DMF), a glyceraldehyde-3-phosphate dehydrogenase (GAPDH) inhibitor, which has been shown to treat autoimmune diseases, was found to promote oxPPP by increasing G6PD expression in tumor cells. In addition, DMF enhanced the efficiency of interleukin-2 (IL-2) therapy while eliminating severe toxicity induced by IL-2 therapy [142]. A recent study has demonstrated that two DNA-binding agents, trabectedin (TRB) and lurbinectedin (LUR), can prompt human macrophages toward a pro-inflammatory state and enhance their anti-tumor activity through metabolic reprogramming, involving ROS production, changes to the mitochondrial inner membrane potential, and the activation of the oxPPP [143].

The development of nano-formulations has increased attention to nano-targeted drug-delivery systems. One study demonstrated the creation of Cu_2_O@Au nanocomposites, which induce oxidative stress pathways, including the PPP. Cu_2_O@Au releases Cu_2_O and Au nanoparticles upon delivery to cancer cells, promoting H_2_O_2_ production and disrupting GSH generation via the PPP. This leads to lipid peroxide accumulation and ferroptosis induction. Ferroptosis triggers immunogenic cell death, dendritic cell maturation, and T cell infiltration and enhances PD-L1 antibody efficacy [135]. Glutaryl-CoA dehydrogenase (GCDH) suppresses HCC by enhancing the crotonylation of key enzymes in the PPP and reducing R5P and lactate production, thereby limiting the Warburg effect. The suppression of the PPP induces oxidative stress and cell senescence, creating an environment antagonistic to tumor growth. The depletion of GCDH renders HCC cells more vulnerable to anti-PD-1 therapy, suggesting that targeting GCDH and the PPP is a promising approach for HCC treatment [144].

### 4.4. Combination Therapy

HIF-1α-induced glucose uptake in PDAC cells enhances both glycolysis and the non-oxPPP, resulting in increased pyrimidine biosynthesis and higher cytoplasmic deoxycytidine triphosphate (dCTP) levels. This elevation in dCTP can diminish gemcitabine’s effectiveness in inhibiting DNA replication in PDAC. In contrast, combining the HIF-1α inhibitors YC1 or digoxin with gemcitabine has been shown to induce apoptosis in PDAC and enhance mouse survival without causing weight loss. Thus, targeting the HIF-1α pathway or pyrimidine biosynthesis may overcome gemcitabine resistance in PDAC and offer a potential combination therapy option with gemcitabine for treating PDAC [92]. Inhibiting G6PD with 6AN has been shown to increase ROS levels, induce cell cycle arrest, and enhance the sensitivity of resistant cells to erlotinib. This highlights the potential of combining 6AN with erlotinib as a promising strategy to address drug resistance in PDAC [78].

Researchers have developed Phy@PLGdH nanosheets, which have been shown to enhance immunogenic cell death, induced by a combination of a 6PGD inhibitor physcion and radiation therapy, thus creating a potent in situ tumor vaccination and amplifying oxidative stress and DNA damage [136]. Furthermore, combining chemotherapy drugs with physcion is more effective in suppressing HCC growth and survival compared to chemotherapy alone [145].

Additionally, the overexpression of LINC01615 competitively binds hnRNPA1 to promote the splicing of *G6PD* precursor mRNA and upregulate G6PD expression. Elevated G6PD increases nucleotide and lipid synthesis, reduces ROS production, and minimizes oxidative damage, thereby promoting cell survival under nutrient starvation and oxaliplatin treatment in CRC, leading to chemoresistance. Therefore, targeting LINC01615 in combination with oxaliplatin treatment is proposed as a potential alternative strategy against chemoresistance in CRC [87]. Another short peptide encoded by lnc-AP, namely, pep-AP, also exhibits potential for similar combination therapy with oxaliplatin. Specifically, pep-AP interacts with the transaldolase (TAL) protein to inhibit its expression. This action weakens the PPP, leading to ROS accumulation and apoptosis, thereby sensitizing CRC to oxaliplatin both in vitro and in vivo [116].

### 4.5. Emerging Therapy

As the next-generation sequencing and omics technologies have advanced and been integrated into medical treatment, the field of gene therapy for cancer has seen remarkable progress over the past few decades. Gene therapy utilizes vectors to introduce genetic material into host cells for therapeutic gene modification [146]. Considering that some key PPP enzymes play an important role in the development of GI cancers, silencing these genes in cancer cells is an attractive treatment option. For example, silencing *G6PD* with lentivirus or non-viral gene delivery vector enhances oxaliplatin anti-tumor effects in CRC xenografts and PDX models [28]. The activation of a tumor suppressor gene like *p53*, *PTEN*, and some ncRNAs that regulate PPP is another promising gene therapy strategy [52,147,148].

To date, a significant proportion of gene therapy research in oncology has been integrated with immunotherapeutic approaches. Among these, chimeric antigen receptor (CAR)-T cell therapy stands out as a promising technique within cancer immunotherapy [149]. In the context of CAR-T cell therapy for cancer, metabolic reprogramming following the pharmacological or genetic ablation of isocitrate dehydrogenase 2 (IDH2) redirects glucose metabolism towards the PPP. This metabolic shift enhances the antioxidative capacity and attenuates CAR-T cell exhaustion, which is particularly beneficial under nutrient-restricted conditions, thereby improving tumor eradication and sustaining CAR-T cell persistence [150].

## 5. Conclusions and Perspectives

Since Horecker et al. fully elucidated the entire PPP in the 1950s, the PPP has been gaining more attention regarding its roles in health and disease [151]. Many studies have shown that cancer cells modulate PPP flux either directly or indirectly to meet their growth demand. The cancer cells need increased PPP flux to generate high NADPH levels to counteract ROS and maintain redox homeostasis. However, ROS can promote tumorigenesis during cancer initiation, but after the tumor is established, it can restrain cancer cell survival and growth. For example, inflammatory bowel disease (IBD) is recognized as a significant risk factor for CRC [152]. TKT ablation in intestinal epithelium leads to extensive mucosal erosion, aberrant tight junctions, impaired barrier function, and increased inflammatory cell infiltration and ROS production in mice, showing similar phenotypes to IBD [153]. In contrast, high expression of TKT promotes the proliferation of CRC cells and increases CRC migration and invasion abilities [39]. The different expression levels of TKT indicate that manipulating PPP enzymes at different stages of GI cancers may have different effects. It is crucial to transition from broad generalizations about the impact of the PPP on cancer development to the time-dependent, location-dependent, and molecule-specific roles that the PPP plays throughout cancer progression.

Over the past decade, significant advancements in the study of metabolic enzymes have revolutionized our comprehension of their roles. Historically, these enzymes were known to catalyze metabolic reactions and were exclusively tied to metabolism-related pathways. Recent insights have expanded this view, revealing a more complex and multifaceted involvement of metabolic enzymes in cellular processes [154]. As mentioned earlier, the nuclear translocation of TKT activates the EGFR pathway, thereby promoting HCC development [61]. Consequently, the non-metabolic function of the PPP enzymes elicited by pathological, especially oncogenic signals or gene mutations, should be taken into consideration.

Considering that the inhibition of the PPP could overcome chemoresistance and radioresistance in GI cancer cells, the synergistic and additive effects of PPP inhibition and other therapeutic strategies are promising. Unfortunately, there are still no specific PPP enzyme inhibitors applied in clinical practice due to their potential toxicity, low efficacy, and off-target effects [155]. Inhibiting the PPP at the organismal level to selectively restrain tumorigenesis without significant adverse physiological consequences remains a challenge. Thus, there is a need for the development of safe and effective PPP inhibitors. Gaining a better understanding of the regulatory mechanisms governing the PPP and the selective advantages it confers to GI cancer cells could unlock novel therapeutic opportunities in GI cancer treatment.

## Figures and Tables

**Figure 1 ijms-26-00610-f001:**
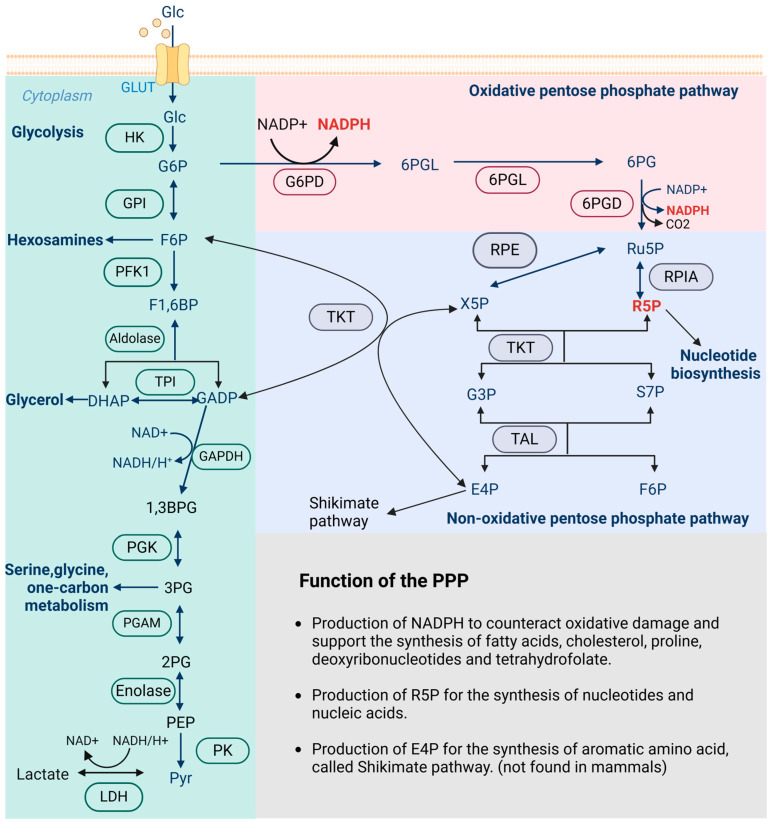
Schematic drawing of the metabolic flux of glycolysis, oxPPP, and non-oxPPP. Abbreviations: 2PG: 2-Phosphoglycerate; 3PG: 3-phosphoglycerate; 6PDGL: 6-phosphogluconolactone; 6PG: 6-phosphogluconate; DHAP: dihydroxyacetone phosphate; E4P: erythrose 4-phosphate; F1,6BP: Fructose-1,6-bisphosphate; F6P: fructose-6-phosphate; G6P: glucose-6-phosphate; GADP: glyceraldehyde 3-phosphate; GLUT: glucose transporter; HK: hexokinase; LDH: lactate dehydrogenase; NAD: nicotinamide adenine dinucleotide; NADPH: nicotinamide adenine dinucleotide phosphate; PEP: phosphoenolpyruvate; PGAM: phosphoglycerate mutase; PFK1: phosphofructokinase-1; R5P: ribose 5-phosphate; Ru5P: ribulose 5-phosphate; S7P: sedoheptulose 7-phosphate; TAL: transaldolase; TKT: transketolase; TPI: triosephosphate isomerase; and Xu5P: xylulose 5-phosphate.

**Figure 2 ijms-26-00610-f002:**
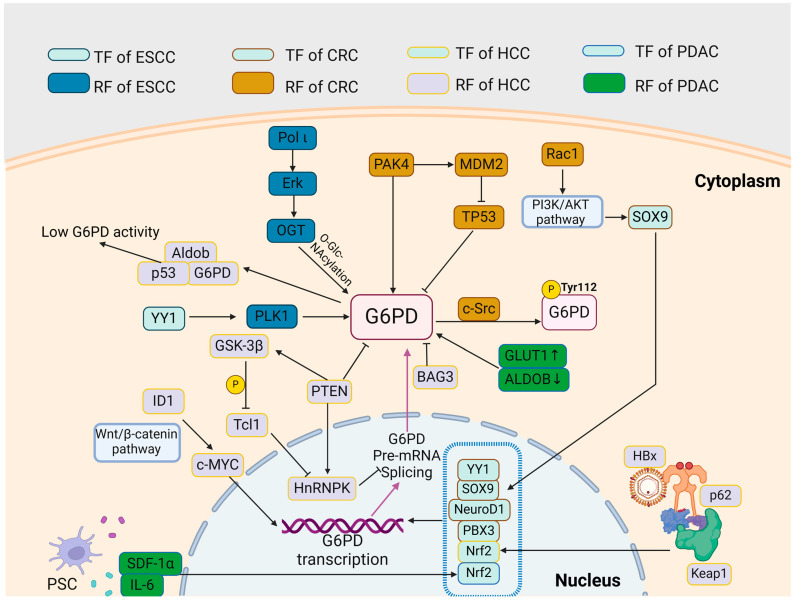
Overview of the regulatory network involving G6PD in GI cancers. Abbreviations: Aldob: aldolase B; BAG3: Bcl-2 associated athanogene 3; CRC: colorectal cancer; ESCC: esophageal squamous cell carcinoma; G6PD: glucose-6-phosphate dehydrogenase; GC: gastric cancer; GSK-3β: glycogen synthase kinase-3β; GLUT1: glucose transporter 1; HBx: hepatitis B virus X protein; HCC: hepatocellular carcinoma; HnRNPK: heterogeneous nuclear ribonuclear protein K; KEAP1: kelch-like ECH-associated protein 1; ID1: inhibitor of differentiation 1; IL-6: interleukin-6; MDM2: murine double minute 2; NeuroD1: neuronal differentiation 1; Nrf2: nuclear factor erythroid 2-related factor 2; OGT: O-GlcNAc transferase; PAK4: p21-activated kinase 4; PBX3: Pre-B-cell leukemia transcription factor 3; PDAC: pancreatic ductal adenocarcinoma; PLK1: polo-like kinase 1; Pol ι: DNA polymerase iota; PSC: pancreatic stellate cell; PTEN: phosphatase and tensin homolog located on chromosome 10; Rac1: ras-related C3 botulinum toxin substrate 1; RF: regulatory factor; SDF-1α: stromal-derived factor-1 alpha; SOX9: sex-determining region Y-box 9; Tcl1: T cell leukemia 1; TF: transcription factor; TP53: tumor protein 53; and YY1: Yin Yang 1.

**Figure 3 ijms-26-00610-f003:**
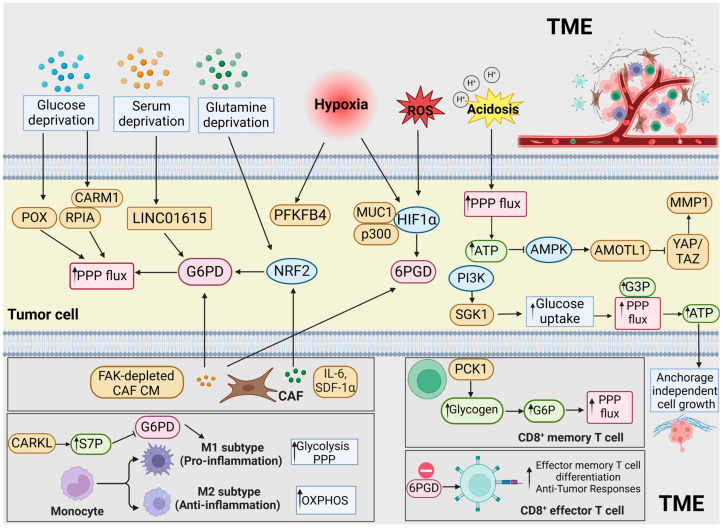
The relationships between the PPP and TME’s elements in GI cancers. Abbreviations: AMPK: AMP-activated protein kinase; AMOTL1: angiomotin like 1; ATP: adenosine triphosphate; CAFs: cancer-associated fibroblasts; CARM1: coactivator-associated arginine methyltransferase 1; CM: conditioned medium; FAK: focal adhesion kinase; G3P: glycerol-3-phosphate; G6P: glucose-6-phosphate; G6PD: glucose-6-phosphate dehydrogenase; HIF1α: hypoxia-inducible factors 1α; IFNγ: interferon-γ; IL-6: interleukin-6; MMP1: matrix metalloproteinase-1; MUC1: mucin 1; NRF2: nuclear factor erythroid 2-related factor 2; OXPHOS: oxidative phosphorylation; PCK1: phosphoenolpyruvate carboxykinase 1; PFKFB4: phosphofructokinase-fructose bisphosphatase 4; PI3K: phosphoinositide 3-kinase; POX: proline oxidase; PPP: pentose phosphate pathway; PFK1: phosphofructokinase-1; RPIA: ribose-5-phosphate isomerase A; ROS: reactive oxygen species; SDF-1α: stromal-derived factor-1α; SGK1: serum and glucocorticoid kinase-1; S7P: sedoheptulose-7-phosphate; TME: tumor microenvironment; TNFα: tumor necrosis factor α; and YAP/TAZ: yes-associated protein and transcriptional coactivator with PDZ-binding motif.

**Table 1 ijms-26-00610-t001:** The regulatory mechanisms of enzymes in the PPP in GI cancers.

GI Cancer Type	Enzyme	Branch of PPP	Regulatory Axis	Functions	Reference
ESCC	G6PD	OxPPP	YY1/↑PLK1/↑G6PD	↓Ferroptosis ↓Chemoradiotherapy sensitivity	[17]
			Pol ι/↑ERK/↑OGT/↑O-GlcNAc of G6PD	↑Proliferation	[18]
	TKT	Non-oxPPP	HMGA1/↑Sp1/↑TKT	↑Proliferation ↑PPP flux ↑NADPH and GSH	[21]
GC	G6PD	OxPPP	Rev-erbα/↓G6PD	↑Proliferation ↑Glycolysis	[23]
CRC	G6PD	OxPPP	PAK4/↑MDM2-mediated p53 ubiquitination/↑G6PD	↑Glucose consumption ↑NADPH production	[29]
			NeuroD1/↑G6PD	↑Proliferation ↓Apoptosis ↑NADPH production ↓ROS level	[30]
			c-Src/↑G6PD Tyr 112 phosphorylation	↑Tumor growth ↑NADPH production ↑Nucleotides synthesis ↑Lipid biosynthesis	[31]
			Rac1/↑PI3K-AKT/↑SOX9/↑G6PD	↑Proliferation ↑Migration ↑Invasion ↑Tumor growth	[32]
			PBX3/↑G6PD	↑Viability ↑Proliferation ↓Apoptosis ↑NADPH production ↓ROS level ↑Lipid biosynthesis ↑Tumor growth	[33]
			YY1/↑G6PD	↑Proliferation ↑Nucleotides synthesis ↑Lipid biosynthesis ↑NADPH production	[36]
	6PGD		ATP13A2/↑TFEB nuclear localization/↑6PGD	↑Proliferation ↑PPP activity ↑Tumor growth	[38]
	TKT	Non-oxPPP	TKT/↑GRP78/↑AKT phosphorylation	↑Proliferation ↑Metastasis ↑Aerobic glycolysis	[39]
	RPIA		Nuclear localization of RPIA/↑β-catenin	↑Proliferation ↑Tumor growth	[44]
			p16/↓mTORC1/↓RPIA	↑Proliferation ↓Senescence ↑Nucleotide synthesis	[43]
			Glucose deprivation/↑CARM1-RPIA interaction/↑RPIA R42 methylation	↑PPP flux ↑ROS clearance ↑Cell growth	[74]
HCC	G6PD	OxPPP	G6PD/↑STAT3 phosphorylation	↑Proliferation ↑Migration ↑Invasion ↑Tumor growth ↑EMT	[48]
			PTEN/↑GSK3β/↓Tcl1/↑hnRNPK/↓G6PD pre-mRNA splicing	↓G6PD dimer formation ↓Proliferation ↑Senescence ↑Sensitivity of HCC to sorafenib	[52]
			Aldob–G6PD–p53 protein complex/↓G6PD activity	↓Tumorigenesis	[53]
			BAG3/↓G6PD	↓Proliferation	[54]
			HBx–p62–Keap1 complex/↑Nrf2/↑G6PD	↑Proliferation	[50]
			ID1/↑Wnt/β-catenin pathway/↑c-MYC/↑G6PD	↑Proliferation ↓Apoptosis ↑Oxaliplatin resistance	[75]
	6PGD		Nrf2/↑6PGD, 6PGD/↓Keap1/↑Nrf2	↑Proliferation ↑Migration	[59]
	TKT	Non-oxPPP	TKT nuclear localization/↑EGFR pathway	↑Proliferation ↑Viability ↑Migration ↑Invasion ↑Metastasis	[61]
			HBx/↑SH2D5/↑interaction of SH2D5 and TKT/↑STAT3 pathway	↑Proliferation ↑Migration ↑Invasion	[63]
			VRK2/↑TKT phosphorylation/↑FBXL6/↑TKT ubiquitination and activation/↑ROS-mTOR axis/↑PD-L1	↑Tumorigenesis ↑Immune evasion ↑Metastasis	[62]
	RPIA		RPIA/↓PP2A activity/↑ERK signaling	↑Proliferation ↑Tumor growth	[76]
PDAC	G6PD	OxPPP	↑GLUT1/↓Aldob/↑G6PD activity	↑Chemoresistance	[66]
			PSC-CM/↑SDF-1α, IL-6/↑Nrf2/↑G6PD	↑Proliferation ↑Glucose metabolism ↑Glutaminolysis ↑Glutathione biosynthesis ↓ROS level	[77]
			ID1/↑c-MYC/↑G6PD	↓Glycolysis ↑PPP flux ↑Erlotinib resistance	[78]
	TKT	Non-oxPPP	↑S100A11 expression/↑H3K4me3 on TKT promoter/↑TKT expression	↑Proliferation ↑Tumor growth	[68]
GIST	6PGD	OxPPP	Long-term imatinib exposure/↑HIF-1α/↑6PGD	↑Proliferation ↓Apoptosis ↑Imatinib-resistant	[72]

“↑” means upregulation and “↓” means downregulation.

## Data Availability

No datasets were generated or analyzed during the current study.

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
