# Peer review of "The Pentose Phosphate Pathway: From Mechanisms to Implications for Gastrointestinal Cancers"

_ijms, 2025, doi:10.3390/ijms26020610_

Round 1
Reviewer 1 Report
Comments and Suggestions for Authors
The authors present a detailed review of the role of the pentose phosphate pathway (PPP) in gastrointestinal (GI) cancers. While the PPP is traditionally known for producing nicotinamide adenine dinucleotide phosphate (NADPH) and ribose-5-phosphate (R5P), this review highlights its broader significance as a key metabolic pathway in cancer. The authors discuss how the PPP contributes to the development and progression of GI cancers and its role in the tumor microenvironment (TME) by maintaining redox balance and providing essential biosynthetic precursors. The review also examines the regulatory mechanisms of PPP enzymes, the pathway's functions within the TME, and potential therapeutic strategies targeting the PPP. This work provides useful insights into the role of the PPP in GI cancer biology and highlights its potential as a target for future treatments.
The work is insightful, well-structured, and written in a clear and engaging manner, making it a valuable contribution to researchers and clinicians working in oncology and related fields.
The only minor remark would concern the clarity of the Figures – the font is too small, please increase the figures, as they are complex, to take all the page, and increase the font to make them readable.
Also, I would suggest to include a chapter on emerging therapies, and describe genotherapy as one of them.
Author Response
We gratefully thank the reviewer for the critical and constructive comments. We have revised our manuscript according to reviewer’s comments. The following is the response to reviewer’s comments on the point-by-point basis. For clarity, our response to reviewer’s comment is highlighted in red. Besides, all the changes made according to reviewers’ suggestion were highlighted in the manuscript.
Reviewer number 1:
Comments 1: The only minor remark would concern the clarity of the Figures – the font is too small, please increase the figures, as they are complex, to take all the page, and increase the font to make them readable.
Response 1: We appreciate reviewer’s comment on the clarity of the figures in our manuscript.
We have made necessary adjustments to ensure that all the figures are of the highest quality and clarity.
We have increased the figures to span the entire page and enlarged the font to make them more readable.
We also uploaded high resolution version of figure files, in case any resolution loss during file-format transforming process. We believe that these changes will significantly enhance the presentation of our work.
Comments 2: Also, I would suggest to include a chapter on emerging therapies, and describe genotherapy as one of them.
Response 2: We have integrated your suggestion into our manuscript and have expanded Section 4.5 to encompass the latest advancements in gene therapy field.
In this updated section, we discuss the remarkable progress in gene therapy for cancer treatment facilitated by next-generation sequencing and omics technologies. We highlight the potential of gene therapy by silencing key PPP enzymes, as well as activating tumor suppressor genes that regulate PPP. Furthermore, we discuss the role of PPP in chimeric antigen receptor (CAR)-T cell therapy. This chapter has been added in Line 565-584 as follows:
“4.5 Emerging therapy
As the next-generation sequencing and omics technologies have advanced and been integrated into medical treatment, the field of gene therapy for cancer has seen remarkable progress over the past few decades. Gene therapy utilizes vectors to introduce genetic material into host cells for therapeutic gene modification[146]. Considering that some key PPP enzymes play an important role in the development of GI cancers, silencing these genes in cancer cells is an attractive treatment option. For example, silencing G6PD with lentivirus or non-viral gene delivery vector enhances oxaliplatin anti-tumor effects in CRC xenografts and PDX models[27]. Besides, activation of tumor suppressor gene like p53, PTEN, and some ncRNAs that regulating PPP is another gene therapy strategy[51, 147, 148].
To date, a significant proportion of gene therapy research in oncology has been integrated with immunotherapeutic approaches. Among these, chimeric antigen receptor (CAR)-T cell therapy stands out as a promising technique within cancer immunotherapy[149]. In the context of CAR-T cell therapy for cancer, the metabolic reprogramming following the pharmacological or genetic ablation of isocitrate dehydrogenase 2 (IDH2) redirects glucose metabolism towards the PPP. This metabolic shift enhances the antioxidative capacity and attenuates CAR-T cell exhaustion, which is particularly beneficial under nutrient-restricted conditions, thereby improving tumor eradication and sustaining CAR-T cell persistence[150].”
We are confident that this addition enriches our manuscript and aligns with the cutting-edge developments in cancer therapy. We appreciate your guidance and look forward to any further comment you may have.
Reviewer 2 Report
Comments and Suggestions for Authors|
Manuscript “The pentose phosphate pathway: from mechanisms to implications in gastrointestinal cancers” presented by Qiao et al. reviews current knowledge on different aspects of the pentose phosphate pathway (PPP) in relation to gastrointestinal cancer development, progression, and treatment. Key PPP molecular players such as enzymes G6PD and TKT are among the main targets of this review. Involvements of extracellular and intracellular PPP- and cancer-related processes are also reviewed. The review is based on a large body of experimental information often presented as a set of not linked facts. The main generalization is done in figures and tables. Fig. 2 and 3 and tables 1 and 2 are the most important parts of the manuscript. Unfortunately, the table 2 is not well described in the text. Therefore, its role is diminished.
Many review papers have been published on similar or identical topics in recent years. Unfortunately, the current manuscript does not mention these works. Here are examples. doi: 10.3892/ol.2019.10112 https://doi.org/10.1159/000519784 https://doi.org/10.1159/000375435 https://doi.org/10.1159/000517771 https://doi.org/10.1007/s13238-014-0082-8 A comparison of references in this and above listed reviews indicates that that this review is biased towards a certain group of authors.
Despite these and listed below drawbacks, this manuscript summarizes some novel findings and can be an important source of information for the scientific community. |
|
Additional comments (Comments are made during continuous reading of the manuscript. Therefore, answers to some raised questions may occur later in the text.) |
|
Abstract 1) “providing biosynthetic precursors” to what? 2) What is “the PPP landscape”? 3) The last sentence should be modified for clarity. |
|
Introduction 1) Additional work on English can be helpful e.g. “facilitates the cellular energy and reducing power generation”. I am not sure that first and second parts of this sentence are well logically connected “As a glucose-oxidizing pathway that runs in parallel to upper glycolysis, the role of pentose phosphate pathway (PPP) in the occurrence and development of GI cancers is receiving more attention.”. Be consistent with abbreviations “(non-oxPPP). The ox-PPP ”. What is the gap in “Thus, it bridges the gap between biosynthetic demands and redox homeostasis,”? “to be mor deeply illustrated.”. “of metabolic flux the glycolysis,”. 2) Not all aerobic glycolysis-based processes can be named as the Warburg effect. The authors stated: “Aerobic glycolysis, a phenomenon termed “Warburg effect”,” e.g. https://cancerandmetabolism.biomedcentral.com/articles/10.1186/2049-3002-2-7 3) Define “the PPP landscape”. 4) Is 3 missing in “6-phosphofructokinase-2/fructose-2,6-bisphosphatase (PFKFB3)” (e.g. phosphofructokinase-2/fructose-2,6-bisphosphatase 3 (PFKFB3))? |
Author Response
We gratefully thank the reviewer for the critical and constructive comments. We have revised our manuscript according to reviewer’s comments. The following is the response to reviewer’s comments on the point-by-point basis. For clarity, our response to reviewer’s comment is highlighted in red. Besides, all the changes made according to reviewers’ suggestion were highlighted in the manuscript.
Reviewer number 2:
Comments 1: The main generalization is done in figures and tables. Fig. 2 and 3 and tables 1 and 2 are the most important parts of the manuscript. Unfortunately, the table 2 is not well described in the text. Therefore, its role is diminished.
Response 1: We have carefully considered your comments regarding the description of table 2 and have taken steps to ensure its role is more prominently featured in the text.
The content of table 2 is referenced partly within the sections "2. Regulatory mechanisms of the PPP in GI cancers" and "3. The landscape of PPP in the TME of GI cancers." To avoid redundancy, we have chosen not to devote a separate paragraph solely to the ncRNAs and PPP.
However, the reason that we placed table 2 under the section 4 is that we believe the ncRNAs listed in table 2 could serve as potential therapeutic targets in GI cancers.
We have reviewed and revised both the table and the figures to enhance their readability and organization. This includes clearer headings, more concise descriptions, and a layout that is more visually coherent with the rest of the manuscript.
We hope that these revisions address your concerns. We appreciate your guidance and are committed to improving the manuscript based on your valuable input.
Comments 2: Many review papers have been published on similar or identical topics in recent years. Unfortunately, the current manuscript does not mention these works. Here are examples.
doi: 10.3892/ol.2019.10112
https://doi.org/10.1159/000519784
https://doi.org/10.1159/000375435
https://doi.org/10.1159/000517771
https://doi.org/10.1007/s13238-014-0082-8
A comparison of references in this and above listed reviews indicates that that this review is biased towards a certain group of authors.”
Response 2: Thank you for your insightful comments and for drawing our attention to the published reviews on similar topics.
We acknowledge that the role of PPP in cancer has been a topic of numerous review articles as you mentioned, reflecting its significance as a research hotspot. Our review, however, specifically focuses on the role of the PPP in GI cancers, which we believe provides a unique point of view within this broader field. In this review, we introduce the role of PPP in initiation and development of GI cancers, and underlined the importance of PPP in GI cancers treatment and prognosis. As a result, our review fits with the topic of special issue “Gastrointestinal Cancer: From Pathophysiology to Novel Therapeutic Approaches”.
We have carefully considered these valuable comments and have now included additional references to the works you've mentioned, specifically: “doi: 10.1159/000519784; doi: 10.1007/s13238-014-0082-8; doi: 10.3892/ol.2019.10112;” after the sentence in the introduction “As a glucose-oxidizing pathway that runs in parallel to upper glycolysis, the role of pentose phosphate pathway (PPP) in the occurrence and development of malignant cancers is receiving more attention[6-8].” (Line 38-40)
Beisdes, we have also identified and included two additional reviews that are highly relevant to our study:
doi: 10.1016/j.tibs.2014.06.005 (Line 40)
doi: 10.1038/s42255-023-00863-2 (Line 47)
We have carefully read these reviews and integrated them into our manuscript where relevant. This has enriched our introduction and provided a more comprehensive view of the current state of research in the field.
Regarding the perceived bias towards a certain group of authors, we would like to clarify that our focus has been on the role of GI cancer in the PPP. This focus has unintentionally led to the inclusion of literature that addresses this specific aspect of the field. Our selection of references is driven by the relevance to our review's topic rather than any preference for specific authors.
We believe that with the inclusion of the additional references, our work has now significant improved to provide a more balanced and comprehensive analysis of the topic.
We appreciate your guidance and hope that these revisions address your concerns. We are committed to maintaining the highest standards of scholarship and scientific rigor in our work.
Additional comments (Comments are made during continuous reading of the manuscript. Therefore, answers to some raised questions may occur later in the text.)
Comments 3: Abstract 1) “providing biosynthetic precursors” to what?
Response 3: Thank you for your comment, we have rephrased “ providing biosynthetic precursors ” as “providing precursors for nucleotide biosynthesis”. (Line 18)
Comments 4: Abstract 2) What is “the PPP landscape”?
Response 4: Thank you for your comment regarding the term "the PPP landscape." In our manuscript, we aim to describe the metabolic landscape of the PPP within tumor cells and the tumor microenvironment (TME). By "PPP landscape," we refer to the dynamic state and features of the PPP in tumor cells and the complex TME, including its metabolites, metabolic enzyme activity, metabolic flux, regulatory axis and crosstalk between cells.
To align with the established terminology and for ease of understanding, we have decided to rephrase "PPP landscape" as "landscape of PPP".
We hope this clarification addresses your question and provides the necessary insight into our use of the term "landscape of PPP".
Comments 5: Abstract 3) The last sentence should be modified for clarity
Response 5: We appreciate your feedback and have further refined the last sentence of the abstract for improved clarity and impact:
“In this review, we summarize the regulatory mechanisms of PPP enzymes, elucidate the landscape of PPP within the TME, and discuss the therapeutic potential of targeting PPP in GI cancers.” (Line 20-22)
We hope this version more effectively captures the scope and significance of our review.
Comments 6: Introduction
1) Additional work on English can be helpful
e.g. “facilitates the cellular energy and reducing power generation”. I am not sure that first and second parts of this sentence are well logically connected.
“As a glucose-oxidizing pathway that runs in parallel to upper glycolysis, the role of pentose phosphate pathway (PPP) in the occurrence and development of GI cancers is receiving more attention.”. Be consistent with abbreviations “(non-oxPPP). The ox-PPP ”. What is the gap in “Thus, it bridges the gap between biosynthetic demands and redox homeostasis,”? “to be mor deeply illustrated.”. “of metabolic flux the glycolysis,”.
Response 6: We appreciate your observation regarding the logical connection in the sentence, " facilitates the cellular energy and reducing power generation." To enhance the clarity and logical flow, we have revised the sentence as follows: "enhances cellular energy production and generates reducing power, "(Line 35-36) .
Upon your suggestion, we have made the necessary amendment to remove the hyphen from "ox-PPP" and now use "oxPPP" throughout the text. This change has been applied consistently wherever the term appears in the manuscript.
Upon your suggestion, we have revisited the sentence "Thus, it bridges the gap between biosynthetic demands and redox homeostasis." We acknowledge that the term "gap" may not accurately convey the intended meaning in this context. We have revised the sentence to more accurately reflect the dual role of the PPP in both biosynthesis and redox processes. The revised sentence is as follows: "Thus, it integrates the biosynthetic demands with the maintenance of redox homeostasis,” (Line 45-46)
In the phrase "to be more deeply illustrated," an "e" was mistakenly omitted after "mor." The correct phrase should read "to be more deeply illustrated". (Line 51)
In the context of "metabolic flux the glycolysis," there was an oversight in not including the preposition "of" after "flux." The corrected phrase should be "of metabolic flux of the glycolysis". (Line 58)
We appreciate your attention to details and have made efforts to ensure that similar errors were promptly corrected in the revised version of our manuscript.
Comments 7: Introduction
2) Not all aerobic glycolysis-based processes can be named as the Warburg effect. The authors stated: “Aerobic glycolysis, a phenomenon termed “Warburg effect”,” e.g. https://cancerandmetabolism.biomedcentral.com/articles/10.1186/2049-3002-2-7
Response 7: Thank you for your comment regarding the use of the term "Warburg effect" in relation to aerobic glycolysis. We agree with reviewer’s comment, the term “Warburg effect” only refers to aerobic glycolysis of cancer cells. We have recognized that our initial statement could lead to misunderstanding and have taken measures to correct it.
The sentence in question has been revised to accurately reflect the definition of the Warburg effect. The updated sentence now reads: " The Warburg effect is defined by the preferential conversion of glucose into lactate, despite the presence of oxygen and functional mitochondria in cultured tumor tissues[4]." (Line 33-35)
We hope that these revisions address your concerns. We are grateful for your guidance and for the opportunity to improve the clarity and accuracy of our work.
Comments 8: Introduction
3) Define “the PPP landscape”.
Response 8: Thank you for your comment regarding the term "the PPP landscape." In our manuscript, we aim to describe the metabolic landscape of the PPP within tumor cells and the tumor microenvironment (TME). By "PPP landscape," we refer to the dynamic state and features of the PPP in tumor cells and the complex TME, including its metabolites, metabolic enzyme activity, metabolic flux, regulatory axis and crosstalk between cells.
To align with the established terminology and for ease of understanding, we have decided to rephrase "PPP landscape" as "landscape of PPP".
We hope this clarification addresses your question and provides the necessary insight into our use of the term "landscape of PPP".
Comments 9: Introduction
4) Is 3 missing in “6-phosphofructokinase-2/fructose-2,6-bisphosphatase (PFKFB3)” (e.g. phosphofructokinase-2/fructose-2,6-bisphosphatase 3 (PFKFB3))?
Response 9: Thank you for your attention to detail. We have made the necessary correction by adding the numeral "3" to the enzyme's name, and it now reads as "phosphofructokinase-2/fructose-2,6-bisphosphatase 3 (PFKFB3)". (Line 101) Additionally, we have conducted a thorough review of the entire manuscript to ensure that all abbreviations are correctly and consistently used throughout the text. We appreciate your feedback and have taken the steps to enhance the clarity and accuracy of our work.
Round 2
Reviewer 2 Report
Comments and Suggestions for Authors
The authors still need to define "landscape of PPP" in the text. Landscape is broadly used word and may cover everything including genes encoding enzymes involved in PPP. Figure 3 even includes cells. What is landscape in this case - everything?
Revision of the Warburg effect was done in the correct direction. However, current version limits it to cultured tumor tissues which is only partially correct. " The Warburg effect is defined by the preferential conversion of glucose into lactate, despite the presence of oxygen and functional mitochondria in cultured tumor tissues[4]."
Additional attention should be paid to the use of “microenvironment”. The immune microenvironment is described with a focus on cells. However, other sections describing the microenvironment tend to evaluate extracellular liquids. Can cells be described as microenvironment?
Author Response
Comments 1: The authors still need to define "landscape of PPP" in the text. Landscape is broadly used word and may cover everything including genes encoding enzymes involved in PPP. Figure 3 even includes cells. What is landscape in this case - everything?
Response 1: Thank you for your comment regarding the clarification needed for the term "landscape of PPP" in our manuscript. We agree with reviewer that ‘landscape’ is too broad to use in our case. After careful discussion, we decided to remove this term and switch the section 3 title to “PPP and the TME in GI cancers”, as highlighted in the revised manuscript. In addition to that, we have replaced the term "landscape" with "relationship" throughout the article to better convey the connection between the PPP and the TME.
Regarding Figure 3, we have replaced its title with "The Relationships between PPP and TME's Elements in GI Cancers." In the upper half of the figure, we describe the impact of nutrient deprivation, hypoxia, and acidic environment in the TME on the PPP of tumor cells. In the lower half, we illustrate how alterations within the PPP of immune cells in the TME may affect their functionality.
Comments 2: Revision of the Warburg effect was done in the correct direction. However, current version limits it to cultured tumor tissues which is only partially correct. " The Warburg effect is defined by the preferential conversion of glucose into lactate, despite the presence of oxygen and functional mitochondria in cultured tumor tissues[4]."
Response 2: Thank you for your valuable feedback. We have revised the definition of the Warburg effect as suggested, expanding it beyond cultured tumor tissues to cancer cells. The updated version now reads: “The Warburg effect refers to the preference of cancer cells to convert glucose to lactate through glycolysis, even in the presence of sufficient oxygen and functional mitochondria, which is closely linked to rapid cancer cell proliferation [4].” (Line 33-36)
Comments 3: Additional attention should be paid to the use of “microenvironment”. The immune microenvironment is described with a focus on cells. However, other sections describing the microenvironment tend to evaluate extracellular liquids. Can cells be described as microenvironment?
Response 3:
Thank you for your observation regarding the use of the term "microenvironment" in our manuscript. We appreciate your comment on the need for clarity in distinguishing between the cellular components and the broader extracellular context that constitutes the microenvironment.
It is important to clarify that the term "tumor microenvironment" refers to the surrounding microenvironment in which tumor cells exist, including surrounding blood vessels, immune cells, fibroblasts, bone marrow-derived inflammatory cells, various signaling molecules and extracellular matrix, which are crucial for regulating tumor growth and metastasis.
As you mentioned, the section on the immune microenvironment in our manuscript primarily focuses on the changes in the PPP within immune cells, thereby affecting their functions in the TME. Therefore, we have revised the title to "PPP and tumor-infiltrating immunocytes". to emphasize that it is the PPP alterations within immune cells which influence the TME.
We hope that these revisions and clarifications address your concerns and provide a more precise understanding of the term "microenvironment" in the context of our review.
